# Can Industrial Co-Agglomeration between Producer Services and Manufacturing Reduce Carbon Intensity in China?

**Tuochen Li, Dongri Han** ***, Shaosong Feng and Lei Liang**

School of Economics and Management, Harbin Engineering University, Harbin 150001, China

* Correspondence: handongri@hrbeu.edu.cn; Tel.: +86-188-4510-5191

**Abstract:** Climate change poses unprecedented challenges for humanity. Reducing carbon intensity is an inevitable choice for tackling climate change and promoting sustainable development. China has made some emission reduction commitments in the international community to promote the decoupling of China's economic development from carbon emissions. The realization of the industrial structure from the "single-wheel drive" of the manufacturing to the "two-wheel drive" economic development model of the service industry and the manufacturing has become a key measure to achieve China's economic intensive development. According to resource misallocation situation in different regions, this paper explored the impact of the collaborative agglomeration between producer services and manufacturing (hereinafter referred to as industrial co-agglomeration) on carbon intensity. The research results show that the carbon intensity is decreasing year by year, and the degree of intensification of China's economic growth continues to increase. Moreover, the effect of industrial co-agglomeration to promote carbon emission reduction is significantly limited by the degree of misallocated resources, and there is a double threshold effect. Specifically, in areas where resource allocation is reasonable, industrial co-agglomeration can produce significant agglomeration effects and promote carbon intensity reduction. Once the degree of misallocated resources exceeds a threshold level, the agglomeration effect will turn into a crowding effect, resulting in an inability to reduce carbon intensity. We comprehensively analyzed the driving factors for reducing carbon intensity and proposed policy pathways for achieving China's carbon intensity target.

**Keywords:** industrial co-agglomeration; carbon intensity; resource misallocation; threshold model; China

## 1. Introduction

Global warming caused by human production and consumption activities is a serious crisis facing the world [1]. To meet this crisis, countries must change the way industrial production and human life work to promote the sustainable development of a low-carbon economy [2–4]. In 2018, China's total carbon dioxide ($CO_2$) emissions exceeded 10 billion tons, exceeding the combined carbon emissions of the United States and Europe, China's economic development direction is of great significance to global climate efforts [3,5]. As a responsible developing country, China put forward its goal at the 2015 Paris Climate Conference: To achieve a peak in $CO_2$ emissions around 2030, with carbon intensity reduced by 60% to 65% compared to 2005. Therefore, under the background of increasingly tight carbon emission reduction constraints, how to reduce carbon emissions while ensuring high-quality economic development is the core issue under the "new normal" [6,7].

To promote the green transformation of industrial structure, based on the actual development of China, strengthen the benign interaction between industries is more effective than simply reducing the

proportion of manufacturing [8]. As an intermediate input industry for industrial development, the productive service industry is characterized by a strong knowledge and strong service [9], and the co-agglomeration (collaborative agglomeration) effect with manufacturing is more obvious [10,11]. Through the industrial co-agglomeration, significant economies of scale can be produced, which is conducive to deepening processing, extending the industrial chain, reducing transaction costs, thereby improving the level of green technology innovation and reducing $CO_2$ emissions [12]. Moreover, in recent years, the Chinese government has paid special attention to the coordinated development of producer services and manufacturing: The 2016 government's work report emphasizes that it is necessary to vigorously develop modern service industry clusters in the next five years and accelerate the transition from productive manufacturing to service-oriented manufacturing. In 2018, the "Central Economic Work Conference" pointed out that China's tertiary industry has great potential for development, we must constantly improve the proportion of the service industry in the national economy and promote industrial integration, especially the productive services and manufacturing industry. Therefore, promote industrial co-agglomeration has become an important step to shift China's economy from extensive growth to green and low-carbon development, and it is also a pivotal move to coordinate the sharp contradiction between ecological protection and economic growth.

Due to a large number of distortions in the operation of China's market economy [13], resource endowments among regions are significantly different [14], and the spillover effects of industrial co-agglomeration is not certain. This is because the industrial co-agglomeration effect is related to the choice of agglomeration location, that is, the threshold effect of development [15]. In resource misallocation areas, industrial co-agglomeration may exacerbate factor market distortions, hinder regional technological progress, and create crowding effects [16]. On the contrary, if the capital and labor factors brought about by industrial agglomeration are integrated with local resource endowments, the industrial structure will be optimized, which will help enterprises concentrate on production, pollution control, management and concentrated consumption of the environment, resulting in positive externalities [8]. Thus, does industrial co-agglomeration promote China's carbon emission reduction? How to optimize China's industrial structure and achieve carbon emission reduction targets? This paper has practical significance for China to transform its economic development model and take a new road to industrialization. At the same time, it also has important reference significance for other developing countries to achieve low-carbon development.

The sections are arranged as follows:

Section 2 presents a comprehensive introduction to this article related literature. In Section 3, first, we calculated the output elasticity of capital and labor (Section 3.1), then we calculated the misallocation index of capital and labor (Section 3.2), on this basis, we measured the misallocation degree of resources (Section 3.3), and has analyzed it comprehensively (Section 3.4). In Section 4, the dependent variable, that is, the carbon intensity, is measured and analyzed. Section 5 presents threshold model and independent variable (industrial co-agglomeration) and control variables, as well as the threshold variable (resource misallocation) and the dependent variable (carbon intensity), which are the results of Sections 3 and 4 above, respectively. Section 6 discusses the results of threshold effect tests, and Section 7 is the conclusions.

## 2. Literature Review

The idea of inter-industry associations leading to industrial co-agglomeration can be traced back to Marshall's (1890) study, which comprehensively analyzes the reasons for industrial agglomeration, arguing that intermediate inputs and shared labor and knowledge spillovers induce firms to gather in a certain area, and get the external economy [17]. On this basis, from the input-output perspective, Venables (1996) proposed a vertical correlation model (CPVL model) to incorporate inter-industry linkages into the industrial co-agglomeration analysis framework [18]. Furthermore, with the increasingly fierce competition between countries and enterprises, an industrial cluster linking production services to manufacturing has been formed, which has promoted the development of

industrial integration. Scholars have analyzed the industrial co-agglomeration of different countries, such as Japan [19], the U.S.A [20], India [21], China [22] and Italy [23].

For the study of the effect of industrial co-agglomeration, there are mainly three aspects.

The first view is that the industrial co-agglomeration has a positive effect. Through a survey of Asian industrial clusters, Low [24] proposed industrial co-agglomeration between producer services and traditional manufacturing, which can generate valuable spillover effects, promote innovation and improve industrial quality, and create sustained growth. Similarly, Wu and Yang [25] also found that the industrial co-agglomeration promotes the improvement of urbanization level, but there are obvious differences between regions, the improvement of urbanization level in the western region of China is most prominent. Lanaspa et al. [26] proposed that industrial co-agglomeration promotes economic growth. Moreover, Connell et al. [27] demonstrated that industrial clusters have significant positive externalities that promote knowledge sharing and collaborative innovation.

Second, the positive effects of industrial co-agglomeration are not significant. Based on panel data from the eastern coastal areas of China, Chen and Chen [28] argued that the industrial co-agglomeration has little effect on the synergistic innovation of enterprises. Zhuang et al. [29] analyzed that the panel data in 31 provinces and cities in China and found that the industrial co-agglomeration inhibits employment growth.

The third view is that the effects of industrial co-agglomeration exhibit nonlinear characteristics. Chen et al. [30] and Ke et al. [10] have deeply deconstructed the industrial co-agglomeration from the perspective of industrial linkage, and they believe that the input-output relationship is the intrinsic reason for the formation of industrial co-agglomeration. However, because there are many types of industries in the producer services, so are the manufacturing industries, therefore, there are not one-way one-to-one pairing relationship, but more of a network-like relationship, which makes the dual gathering in the same city produces a dual effect, namely the complementary effect and the crowding-out effect, in this way, the industrial co-agglomeration effect shows the "inverted U" type of trajectory. Dou and Liu [31] conducted a related research on the construction of threshold regression models from 285 prefecture-level cities in China from 2003 to 2012, Zhou and Chen [32] took the Changsha-Zhuzhou-Xiangtan urban agglomeration as research object, Wang [33] has studied the five large cities, all of them supported the above points.

Previous research on resource misallocation has focused on the following two areas:

First, discuss the causes and consequences of resource misallocation. Most scholars believe that financial market friction is the main cause of resource misallocation [34]. By comparing the financing costs of different companies entering the capital market, Gilchrist et al. [35] confirmed that financial friction has significant negative externalities, which is not conducive to resource optimization. Similarly, Sandleris and Wright [36] proposed that the financial crisis has seriously worsened the distribution of resources both across and within sectors in Argentina, and reduced total factor productivity by 10%. Other factors also lead to resource misallocation, such as trade barriers [37], credit market imperfections [38], etc. In-depth analysis of the mystery of low productivity growth in southern European countries, Gopinath et al. [39] found that capital misallocation seriously hindered economic development and inhibited productivity growth. Furthermore, Calligaris et al. [40] also proposed that resource misallocation hinders productivity. Han and Zhang [41] put forward that excessive government intervention is an important reason for resource misallocation. Moreover, Jin et al. [42] put forward that the difference in ownership is one of the reasons for resource misallocation.

Second, discuss the efficiency improvement brought by correcting resource misallocation. Through an empirical analysis of Portuguese enterprise-level data, Dias et al. [43] found that resource misallocation caused Portugal's annual GDP growth rate to decrease significantly, from 1996 to 2011, by about 1.3 percentage points. At the same time, resource misallocation has not improved, and has become more serious over the years. Thus, improving resource misallocation can greatly promote the economic development of Portugal. Similarly, Buera et al. [44] proposed the large-scale economic reform can remove resources distortions, and promote TFP rises. By constructing a macro

model of monopolistic competition, Hsieh and Klenow [13] empirically found that if China's resource allocation efficiency reaches the US level, the total TFP will increase by 30–50%. Moreover, based on the measurement of Hsieh and Klenow, Ye and Lou [45] proposed that if the level of resource misallocation is reduced by one unit, the average consumption demand will increase by 0.62 percentage points, and the excessive investment demand will be alleviated by 6.84 percentage points. Through data analysis at the enterprise level in different countries, Bartelsman et al. [46] proposed that improving resource misallocation is an effective way to improve the level of national economic development, and predicted that if resource misallocation between institutions could be eliminated, the national economic output would be greatly increased, and some countries could achieve as high as 15%. Luo et al. [47] believe that correcting resource misallocation can increase China's per capita GDP by 115.6%. It is relevant to note that, due to the neglecting or mis-measuring adjustment costs, neglecting differences in technologies across establishments, etc., it may lead to an excessive exaggeration of the misallocation degree of resources [48]. The above research basically affirms that gap between China and developed countries is widening because of resource misallocation, and is an important constraint to the high-quality development of China's economy [13,48,49].

From the above-related research results, first, in the past, most of the research industry agglomeration were concentrated on the issue of single industry agglomeration, and scholars did not pay enough attention to the industrial co-agglomeration, especially the productive services and manufacturing industry. Second, there is no consensus on the relationship between industrial co-agglomeration and carbon intensity. Third, previous studies have focused on the allocation of resources in various countries, while there is less literature about the allocation of resources within emerging economies, such as regions of China. Fourthly, the previous research on the relationship between collaborative aggregation, resource misallocation and carbon emissions intensity mainly focused on the analysis of the relationship between two parties, ignoring the non-linear relationship between three parties. To overcome the above-mentioned deficiencies, based on the threshold model, we empirically analyze the nonlinear relationship between the three, clarify the effect of industrial co-agglomeration on carbon intensity, and realize the decouple economic development from carbon emissions.

## 3. Measuring the Regional Misallocation Degree of Resources

### 3.1. Calculating the Output Elasticity of Resource

To measure the misallocation degree of resources, that is, labor and capital among regions, first, we need to measure the output elasticity of labor and capital. This paper refers to the practice of Osotimehin [50], using the Solow residual value to measure the elasticity. Suppose the production function is a C-D production function with the same scale return, i.e.,

$$Y_{it} = A K_{it}^{\beta_1} L_{it}^{\beta_2}, \tag{1}$$

where $\beta_1 + \beta_2 = 1$. The logarithm is taken on both sides of the function, and the individual effect $\mu_i$ and the time effect $v_t$ are added to the model. The specific form is as follows:

$$ln(Y_{it}/L_{it}) = lnA + \beta_1 ln(K_{it}/L_{it}) + \mu_i + v_t + \varepsilon_{it}, \tag{2}$$

where $Y_{it}$ denotes the output variable, which is expressed by the GDP of each region, and based on 2009, the industrial product price index is used to calculate the income of other years as constant price. $K_{it}$ denotes the R&D (research and development) capital input, that is, the stock of fixed capital in each region in this paper, using the perpetual inventory method for stock accounting. $L_{it}$ denotes the labor input, use the average annual employment in each region to express.

On this basis, this paper estimates the capital and labor output elasticity of each province. Due to the heterogeneity of each province, the factor output elasticity of each region may be different. Therefore,

the variable coefficient panel model with variable intercept and the variable slope is suitable. We use the least squares dummy variable method to estimate the elasticity of factors output in each region. The variable coefficient model is estimated by introducing an interaction term between the dummy variable and the explanatory variable of the variable coefficient ($ln(K_{it}/L_{it})$) in the regression equation, so that the individual sections can have different estimation coefficients. Through calculation, we can get the output elasticity of capital and labor among regions.

### 3.2. Calculating the Misallocation Index of Capital and Labor

Under competitive equilibrium, according to the method of Hsieh and Klenow [13], we can define two types of "distortion coefficients" as follows:

$$\gamma_{Ki} = \frac{1}{1+\tau_{Ki}}, \gamma_{Li} = \frac{1}{1+\tau_{Li}}, \tag{3}$$

where the $\gamma_{Ki}$ and $\gamma_{Li}$ represent the absolute distortion factor of the elements price, which indicates the addition of resources when there is no distortion. The distortion coefficient reflects the degree of deviation of the actual resource allocation from the theoretical effective resource allocation. When the ratio is less than 1, it indicates that resource cost in the region is high and the configuration is insufficient. Because when deciding the allocation of elements among regions, what is important is the "relative" rather than "absolute" distortion of the factor price [13]. Therefore, in the actual calculation, the relative distortion coefficient is used instead of the absolute distortion coefficient:

$$\hat{\gamma}_{Ki} = \frac{\gamma_{Ki}}{\sum_{i=1}^{n}(\frac{s_{it}\beta_{Ki}}{\beta_K})\tau_{Ki}}, \hat{\gamma}_{Li} = \frac{\gamma_{Li}}{\sum_{i=1}^{n}(\frac{s_{it}\beta_{Li}}{\beta_L})\tau_{Li}} \tag{4}$$

Transforming the formula (4), we can get:

$$\hat{\gamma}_{Ki} = \frac{K_i}{K}\left|\frac{s_i\beta_{Ki}}{\beta_K}, \hat{\gamma}_{Hi} = \frac{L_i}{L}\right|\frac{s_i\beta_{Li}}{\beta_L}, \tag{5}$$

where $\frac{K_i}{K}$ and $\frac{L_i}{L}$ represent the actual proportion of capital invested in total capital and the labor invested in total labor, of province $i$ for year $t$. $s_i = \frac{p_i y_i}{Y}$ denotes the actual proportion of output in national output, of province $i$ for year $t$. $\beta_K = \sum_{i=1}^{n}s_i\beta_{Ki}$ and $\beta_L = \sum_{i=1}^{n}s_i\beta_{Li}$ represent the output- weighted capital contribution value and labor contribution value, $\frac{s_i\beta_i}{\beta}$ represents the theoretical ratio of factor that needs to be invested in the effective allocation of resource.

### 3.3. Calculating the Misallocation Degree of Resources

Based on the misallocation indexes calculated above, refer to the method of Bai and Liu [51], we can obtain the misallocation degree of resources among regions by taking the average. Due to there are two cases of the insufficient configuration and excessive configuration, in order to make the regression direction consistent, this paper refers to the practice of Ji et al. [52], and performs the absolute value processing on the degree of misallocation. The larger the value, the more serious resource misallocation.

### 3.4. Analysis of the Misallocation Degree of Resources among Regions

According to the above method, the regional misallocation degree of resources in 2009–2016 is measured. Taking 2009 and 2016 as examples, we explain the misallocation degree of resources with a spatial distribution map (Figure 1). From the perspective of regional distribution, resource misallocation exists in each region of China, and there is a large difference between different regions.

In 2009, the allocation of factors in most parts of China was not ideal. At that time, the economic development foundation of the eastern region was relatively strong. With its unique geographical

advantages, the competitive effect caused by the survival of the fittest made resources significantly inclined to flow to high-efficiency enterprises, and a large number of capital and labor gathering to the eastern region. However, in the short term, the market cannot fully absorb these factors, thus causing certain mismatches. Instead, the economic base of the central and western regions is weak, development is limited, capital and labor are lost, and the elements required for economic development are seriously inadequate, thus causing certain mismatches. Afterwards, in order to promote the balanced development of China's region, the government promulgated a series of policies, such as "Western Development" and "Northeast Revitalization Strategy", to promote the transfer of industrial factor resources to underdeveloped regions. Moreover, China's market economy is developing more and more fully. Therefore, by 2016, China's resource misallocation situation has been improved to some extent. However, we also found that the misallocation degree of resources in Jiangsu and Beijing provinces is still serious, and the misallocation degree of resources in Qinghai has become more serious, all of which require special attention.

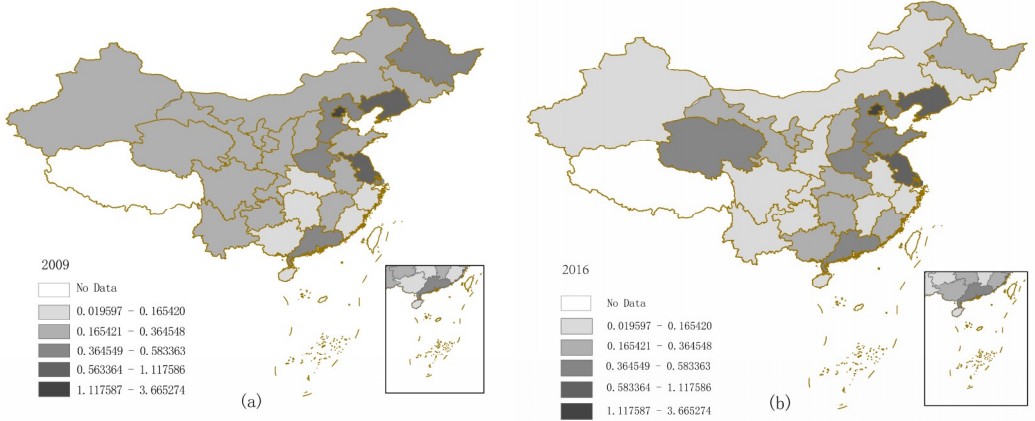

**Figure 1.** The misallocation degree of resources in China (2009 and 2016).

## 4. Measuring the Carbon intensity

### 4.1. Calculating the Carbon Intensity

Carbon intensity is $CO_2$ emissions per unit GDP [5,53], in order to calculate the carbon intensity; first, it is necessary to measure $CO_2$ emissions accurately. According to IPCC (2006) and the Office of the National Climate Change Coordination Group, the burning and using of fossil fuels is the main cause of the increase in global $CO_2$ emissions, at the same time, non-fossil fuels, such as cement, lime, calcium carbide, and steel, will also emit $CO_2$, due to physical and chemical reactions during industrial production, which is often ignored in the calculation of carbon emissions [54]. Among the $CO_2$ emissions from all industrial processes, cement accounts for 56.8%, lime accounts for 33.7%, and calcium carbide and steel production accounts for less than 10% [55]. Since the data of lime, calcium carbide, and steel are difficult to obtain and the proportion of emissions is relatively small, this paper is not included in the calculation. Based on energy consumption, we measure the total amount of CO2 emissions according to the method provided by IPCC, including seven major fossil energy sources (coal, coke, gasoline, diesel, fuel, natural gas, kerosene), and non-fossil energy (cement). The calculation formula is as follows:

$$EC = \sum_{j=1}^{7} C_j = \frac{44}{12} \sum_{j=1}^{7} E_j \times CF_j \times CC_j \times COF_j = \sum_{i=1}^{7} \frac{44}{12} \alpha_j E_j, \tag{6}$$

where EC represents carbon emissions of fossil fuel consumption; *j* represents the fossil fuel category; *E* represents fossil fuel consumption; *CF* represents low calorific value; *CC* represents carbon content;

*COF* represents the rate of carbon oxidation; $\alpha = CF \times CC \times COF$ denotes the carbon emissions coefficients from the IPCC.

$$CC = Q \times \beta, \tag{7}$$

where *CC* represents carbon emissions of cement consumption; *Q* denotes total cement production; *β* denotes the carbon emissions coefficients, refer to Zhang H [50,55], the value is $0.5270 tCO_2/t$.

Hence, the formula for calculating the total amount of $CO_2$ emissions in each region is $CO_2 = EC + CC$.

*4.2. Analysis of the Carbon Intensity*

Overall, China's carbon intensity was decreasing from 2009 to 2016 (Figure 2), indicating that China's economic growth was becoming increasingly intensive.

From the perspective of regional distribution, the carbon intensity of provinces in the eastern region is the lowest, such as Beijing, Shanghai and Guangzhou, etc., while the index in other areas is higher, which is consistent with economic development. On account of the strong economic strength and high innovation level in the eastern region, it is actively promoted in terms of orderly development of resources, intensive use, directly promoting the improvement of the regional ecological environment. The central and western regions are the latecomers of China's economic development. The economic foundation is weak, scientific research capabilities are limited, and the development of high-tech industries is insufficient. In addition, the transfer of some extensive domestic and international industries has been carried out, resulting in the deterioration of the ecological environment in the region. In order to balance regional development, a large amount of supports provided by the Chinese government for economically underdeveloped regions recently, which has improved the economic situation there.

It is worth noting that the carbon intensity of Shanxi Province is significantly higher than that of other provinces. The reason is mainly because Shanxi has a large amount of coal resources and is the main coal supply area in China. The economic development of the region mainly depends on coal production capacity and belongs to the extensive economic growth mode, resulting in long-term high carbon intensity.

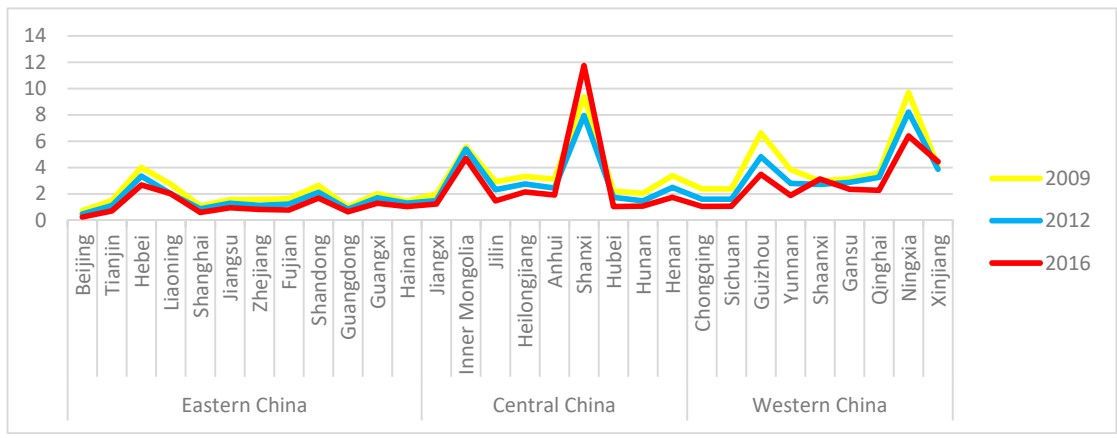

**Figure 2.** Carbon intensity in China (2009, 2012, 2016).

## 5. Empirical Model and Variables

*5.1. Specifications of the Threshold Model*

From Section 3, which examines the measurement of resource misallocation in various regions of China, it can be seen that the resource allocation situation in China is significantly different. Combined with the theoretical framework of the industrial synergy mechanism, we believe that when

studying its carbon emission reduction effect, regional resource misallocation should be introduced into empirical analysis, not just a linear effect. We further speculate that the relationship between industrial co-agglomeration and carbon intensity may have a typical "threshold effect", that is, in the case of different economic parameters, the influence of independent variable on a dependent variable may change. In this paper, when resource misallocation break through a certain threshold, the marginal coefficient of influence of industrial co-agglomeration on carbon intensity will become bigger or smaller, even the direction of action will be reversed.

Hansen's nonlinear panel threshold regression model is a statistical analysis method that deals with nonlinear structural mutations. It can identify the data characteristics of unknown variables from the perspective of mathematical statistics, avoiding the bias caused by artificially dividing the threshold variable interval. Scientifically, a significant test was performed on the endogenous threshold effect [56]. The empirical analysis of this method can not only clarify whether industrial co-agglomeration can help achieve carbon emission reduction targets, but also analysis the green energy conservation relationship dominated by industrial co-agglomeration based on the heterogeneity characteristics of different regions in China, that is, the misallocation degree of resources.

Following the knowledge production function framework proposed by Jaffee [57], we used the Hansen's threshold model to empirical analysis. According to the number of thresholds, the threshold regression model can be divided into a single-threshold and multiple-threshold models. Single-threshold and double-threshold models are commonly used; we built a threshold model as follows:

$$CO_{2_{it}} = \theta + \alpha_1 GOV_{it} + \alpha_2 MAR_{it} + \alpha_3 KNO_{it} + \alpha_4 R\&D_{it} + \beta_1 Coagglo_{it} I(\tau \leq \eta) + \\ \beta_2 Coagglo_{it} I(\tau > \eta)_i + u_i + \varepsilon_{it},$$ (8)

where $CO_{2_{it}}$ represents carbon intensity, of province $i$ for year $t$. $Coagglo_{it}$ is the industrial co-agglomeration, of province $i$ for year $t$, $I()$ is the indicator function, $\tau$ is threshold variable (here, resource misallocation). Control variables include: Government support (*GOV*), technology market maturity (*MAR*), knowledge spillover (*KNO*), research and development input (*R&D*). $\eta$ denotes the threshold variable value, $u_i$ denotes the specific effect of the individual, and $\varepsilon_{it}$ is a random disturbance variable.

Then, we build a double threshold model as follows:

$$CO_{2_{it}} = \theta + \alpha_1 GOV_{it} + \alpha_2 MAR_{it} + \alpha_3 KNO_{it} + \alpha_4 R\&D_{it} + \beta_1 Coagglo_{it} I(\tau \leq \eta_1) + \\ \beta_2 Coagglo_{it} I(\eta_1 < \tau \leq \eta_2)_i + \beta_3 Coagglo_{it} I(\tau > \eta_2) u_i + \varepsilon_{it}.$$ (9)

*5.2. Variables and Data Sources*

In order to conduct empirical analysis, it is necessary to introduce variables and data sources of this paper. Resource misallocation and carbon intensity are the results of Sections 3 and 4 above.

In order to accurately measure the industrial co-agglomeration between producer services and manufactures, we use location entropy to measure the producer services industry agglomeration (*Psagglo*) and manufactures agglomeration (*Magglo*) index of various provinces in China [58]. On this basis, through the relative of industrial agglomeration differences measure the industrial co-agglomeration variable (*Coagglo*) [59]. According to the statistics of employment in the inter-provincial industry in China, and referring to the practices of Zhong and Yan [60], Yu [61], etc. This paper chooses to add seven industries as a producer services industry, including transportation, warehousing and postal services; information transmission, computer services and software industry; scientific research, technical services and geological exploration; financial industry; lease and commercial services; wholesale, retail and trade; water environment and public facilities management. The calculation formula is:

$$Coagglo_{it} = 1 - \frac{|Magglo_{it} - Psagglo_{it}|}{Magglo_{it} + Psagglo_{it}}.$$ (10)

There are four control variables. First, government support is conducive to promoting sustainable economic development in various regions. The government's support for sustainable economic activities in the region is mainly reflected in the implementation of grants or tax incentives, which can reduce the cost of corporate green emission reduction activities, stimulate the enthusiasm of enterprises to adopt low-carbon technologies, and have a positive impact on carbon emissions reduction [6,53]. We choose the ratio of local fiscal expenditure to GDP as a proxy variable (*GOV*). Second, the technology market is the medium for knowledge, technology sharing and optimal configuration, and it provides a good communication platform for the flow of low-carbon technology. In general, mature regional technology markets can spread frontier technologies to a large extent, driving low-carbon development of regional enterprises [60,62]. In this paper, the technology market turnover was used to indicate the maturity of the technology market (*MAR*). Third, knowledge spillover is an important feature of industrial agglomeration, and the university is regarded as an important source of local knowledge spillover, due to its clear knowledge production and diffusion function. Knowledge spillover through colleges and universities in various regions is a localized indirect knowledge transfer mechanism [62]. In this paper, the number of regional colleges and universities was used to indicate the knowledge spillover (*KNO*). Finally, R&D human resources are the knowledge reserve of the enterprise, laying a foundation for enterprises to choose the path of intensive economic development [5,63]. We use the "full-time R&D personnel" to measure research and development input (*R&D*).

We take a panel dataset of annual data on 30 provinces in China over 2009–2016 as our research sample. All of the original data were obtained from the National Bureau of Statistics of China. Furthermore, we have processed the data accordingly to eliminate the influence of price factors or dimensional factors. Table 1 summarizes the descriptive statistics of variables.

**Table 1.** Descriptive statistics of variables.

| Variable | Mean | S.D. | Variance | Min | Max |
|---|---|---|---|---|---|
| $CO_2$ | 2.610 | 2.196 | 4.823 | 0.241 | 12.781 |
| $\tau$ | 0.485 | 0.704 | 0.496 | 0.005 | 4.996 |
| *Coagglo* | 0.797 | 0.151 | 0.023 | 0.278 | 1.000 |
| *GOV* | 0.237 | 0.101 | 0.010 | 0.096 | 0.639 |
| *MAR* | 13.278 | 1.735 | 3.009 | 8.623 | 17.490 |
| *KNO* | 4.250 | 0.649 | 0.422 | 2.197 | 5.112 |
| *R&D* | 8.701 | 1.772 | 3.139 | 3.100 | 12.321 |

## 6. Results of Threshold Effect Tests

### 6.1. Threshold Significance Test and Confidence Interval

For the nonlinear threshold model, using the Hansen's model estimation and testing methods [56], first, we need to check whether the threshold effect exists, in this paper, that is, resource misallocation. Then, the number of thresholds needs to be determined. The report is shown in Table 2. We found that the single threshold and the double threshold were significant at the 1% and 5% levels, respectively, while the triple threshold model was not significant (*p*-value = 0.1867). Thus, the double threshold effect is selected below for analysis and discussion.

**Table 2.** Test results of threshold significance.

| Model | F-Value | *p*-Value | Critical Value | | |
|---|---|---|---|---|---|
| | | | 1% | 5% | 10% |
| Single threshold | 151.53 * | 0.0000 | 67.0181 | 35.7013 | 30.2341 |
| Double threshold | 49.55 ** | 0.0033 | 43.8863 | 32.6541 | 27.5792 |
| Triple threshold | 29.27 | 0.1867 | 91.9739 | 61.9715 | 43.4545 |

Note: The *p*-value and the critical value are obtained from 300 bootstrap replications. ***, **, * denote significant levels at 1%, 5%, and 10%, respectively.

Table 3 reports the double threshold values, that are, 0.4320 and 0.6063, and the 95% confidence intervals, that are, [0.4288, 0.4344] and [0.5992, 0.6213].

**Table 3.** Threshold values and confidence intervals.

| Model | Threshold Estimators | 95% Confidence Intervals |
|---|---|---|
| Single threshold | 0.4320 | [0.4288, 0.4344] |
| Double threshold | 0.6063 | [0.5922, 0.6213] |

After the significance test of the threshold effect, the likelihood ratio $LR_n(\gamma)$ was used to construct "non-rejection region", that is, the valid confidence interval of $\gamma$. The "non-rejection region" at the confidence level of $1 - \alpha$ is a series of $\gamma$ values that belong to $LR_n(\gamma) \le c(\alpha)$. When $LR_n(\gamma) \le c(\alpha) = -2ln(1 - \sqrt{1 - \alpha})$, we cannot reject $H_0 : \gamma = \gamma_0$ (Figure 3).

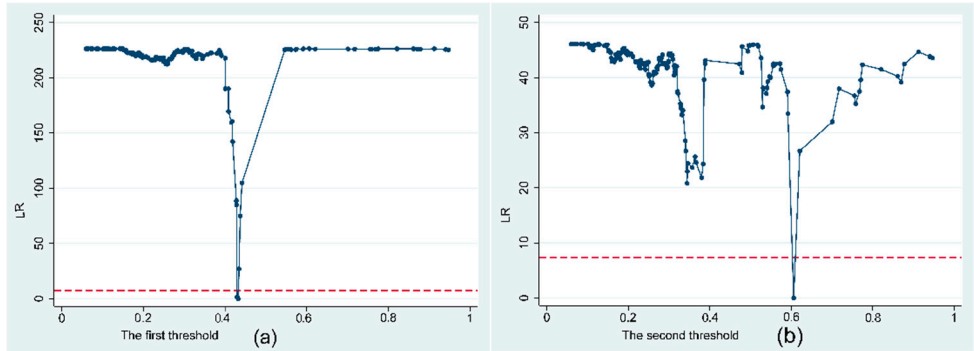

**Figure 3.** The construction of confidence intervals. (**a**) The first threshold value. (**b**) The second threshold value.

### 6.2. Estimation Results of the Threshold Model

According to the above analysis, based on the misallocation degree of resources, there is a double threshold between industrial co-agglomeration and carbon intensity. Next, we further analyze the partition coefficient of each interval.

As shown in Table 4, when resource allocation is reasonable ($\tau \le 0.4320$), the industrial co-agglomeration negatively affects carbon intensity at a significant level of 5%, which means promote industrial co-agglomeration can effectively reduce regional carbon intensity. As the misallocation degree of resources is strengthened ($0.4320 < \tau \le 0.6063$), the mechanism differs, the effect of industrial co-agglomeration on carbon intensity has changed from negative to positive at a significant level of 10%, which means that industrial co-agglomeration gathers to aggravate carbon intensity. However, when the misallocation degree of resources exceeds 0.6063 (seriously resource misallocation), the effect on carbon intensity is not significant. Therefore, it can be found that the industrial co-agglomeration and carbon intensity are not a simple linear relationship, but exist threshold effect of resource misallocation in China. In the lower resource misallocation areas, the positive externalities of industrial co-agglomeration are stronger, which is more conducive to reducing carbon intensity.

In terms of control variables, *GOV* is an advantageous measure to reduce carbon intensity, along with the increasing pressure of international low-carbon, and the practical needs of economic green transition, China's regional governments have shown strong willingness to protect the environment, and took measures, such as financial support and environmental regulations to reduce carbon intensity. *MAR* and *KNO* an also promote carbon intensity reduction, showing that mature technology market and knowledge spillovers can drive enterprises to use new technologies and promote green innovation in the region. In addition, *R&D* has no significant effect on carbon intensity.

**Table 4.** Estimation results of model parameters.

| $CO_2$ | Coef. | Std. Err | t | $p > |t|$ | 95% Conf. Interval | |
|---|---|---|---|---|---|---|
| *GOV* | −3.678 | 1.693 | −2.17 | 0.031 | −7.017 | −0.339 |
| *MAR* | −0.120 | 0.063 | −1.91 | 0.058 | −0.243 | 0.004 |
| *KNO* | −4.678 | 0.652 | −7.17 | 0.000 | −5.964 | −3.392 |
| *R&D* | −0.134 | 0.089 | −1.50 | 0.136 | −0.310 | 0.043 |
| $\tau \leq 0.4320$ | −1.596 | 0.608 | −2.62 | 0.009 | −2.796 | −0.396 |
| $0.4320 < \tau \leq 0.6063$ | 1.389 | 0.618 | 2.25 | 0.026 | 0.169 | 2.608 |
| $\tau > 0.6063$ | 0.082 | 0.706 | 0.12 | 0.907 | −1.310 | 1.475 |
| cons | 26.827 | 2.568 | 10.45 | 0.000 | 21.764 | 31.890 |

*6.3. Discussion*

In general, industrial co-agglomeration can significantly reduce regional carbon intensity. However, we find that industrial co-agglomeration is not a linear effect on carbon intensity, but a nonlinear complex effect with resource misallocation as the threshold. The level of regional sustainable development depends largely on the allocation of factors in the region, making regional carbon intensity more sensitive to resource misallocation conditions, the industry industrial co-agglomeration promotes a large number of resource elements concentrated in a certain area, resulting in "aggregation effect" and "crowding effect", neglecting the allocation of resources in the region, blindly promoting the coordinated industrial co-agglomeration, cannot fundamentally improve the mode of economic development, and reduce carbon intensity. For a long time, China's economy has achieved considerable development relying on factor-driven development, which has caused over-allocation of factors in some regions, and environmental carrying capacity is approaching the limit. In economically underdeveloped areas, due to insufficient resource allocation, they are unable to effectively attract the transfer of factors, resulting in poor economic development in these areas.

Low misallocation degree of resources will effectively reduce the carbon intensity by promoting industrial co-agglomeration, because the factor resources brought by industrial co-agglomeration can be integrated successfully with the resources of the region, and the supply and demand structure of the elements can be reasonably matched to promote industrial structure optimization. Through the producer services are interconnected with the upstream and downstream production links of the manufactures, resources can be recycled, which in turn promotes low-carbon production and green technology innovation, and reduces pollution emissions per unit of output. At the same time, as a high-level factor in the development of manufacturing, the producer services are constantly gathering human capital, intellectual capital, and technology capital. By generating competitive effects, knowledge spillover effects, and economies of scale, it will greatly promote regional green production efficiency, to a certain extent, reduce the pollution paradise effect and achieve green-intensive development.

When the misallocation degree of resources is high, industrial co-agglomeration in the short term further increase $CO_2$ emissions. As economic density and population density continue to rise, land prices, housing rents, operating costs and the consumption of scarce resources are exhausted, putting a burden on the normal living environment, which leads to negative external effects, such as congestion and environmental pollution. In turn, it causes regional ecological efficiency loss and carbon emissions to increase. In addition, when companies use mismatched factor resources for production, they will greatly reduce the economic growth effect, and innovation-driven role brought about by industrial co-agglomeration. In particular, for economically under-developed regions, blindly promoting industrial co-agglomeration will result in pollution superposition. Faced with high environmental costs, companies often produce short-sighted behavior and increase carbon emissions.

All in all, when the misallocation degree of resources exceeds the threshold, the effect of industrial co-agglomeration will change from agglomeration effect to crowing effect, and the impact on carbon intensity will be changed from decreasing to increasing, which will affect the development process of China's green development.

## 7. Conclusions

Improving resource allocation efficiency is an important issue in deepening supply-side structural reforms, reducing inefficient resource inputs, and promoting economic development to high quality. Achieving carbon intensity targets by industrial co-agglomeration is an important step to promote China's green economy, which also has uncertainty and high complexity. Based on the macro environment of international carbon emissions reduction and the "new normal" development stage of the domestic economy, a nonlinear threshold model of industrial co-agglomeration, resource misallocation, and carbon intensity constructed and clarified. Finally, it provides insights on achieving carbon intensity targets.

On the whole, China's economic development model has been increasingly intensive. Under the impetus of a series of policy measures, carbon emissions reduction has achieved certain results. Nevertheless, the process of intensive economic development presents regional fluctuation features and fails to form a unified trend. Evidently, the transformation effects of Shanxi, Shaanxi and Xinjiang are worthy of attention, because the carbon intensity of these three cities has not decreased sufficiently in recent years.

There is a non-line relationship between industrial co-agglomeration and carbon intensity based on resource misallocation. If the regional resources are properly allocated, through industrial co-agglomeration, it will successfully promote low-carbon development in the region, thereby reducing carbon intensity. Once the misallocation degree of resources exceeds the threshold, the crowding effect of industrial co-agglomeration will appear and increase the carbon intensity.

According to the research conclusions, in order to further optimize resource allocation and enhance industrial co-agglomeration to drive carbon emissions reduction, we propose the following recommendations:

The first is to promote supply-side reform, uphold the development strategy of production service industry and manufacturing "two-wheel" drive, to rely on the market's "invisible hand" and the government's "tangible hand" in strengthening the integration of the two, leading the manufacturing to promote high-end in the global value chain, extending the industrial chain along the production service industry at both ends of the "smile curve". Meanwhile, transforming the redundant organizational structure of manufacturing enterprises, while actively developing the manufacturing, moderately lowering the threshold for the service industry and attracting the entry of related production services. Build a cross-border industrial co-agglomeration model, foster a multi-functional industrial cluster that combines manufacturing and service, take the agglomeration effect of industrial co-agglomeration and reduce regional carbon intensity.

Second, the government should deepen market reforms, reduce the direct allocation of economic resources, create a free market competition environment, and optimize the allocation of resources by market mechanisms. Specifically, the central government should break the administrative monopoly, make the market play a fundamental role, continue to promote decentralization, and maximize the innovation vitality of enterprises. Local governments need to plan the allocation of resources within the province, rely on their own advantages, reduce local trade barriers, and ensure the barrier-free circulation of labor, capital and other resources. In addition, in the context of economic globalization, it is necessary to encourage companies to actively participate in the international division of labor, configuring resources on a global scale will help improve China's resource misallocation.

Third, considering the heterogeneity threshold effect of resource misallocation, it should be adapted to local conditions and coordinate regional development. According to their different levels of resource misallocation, each region chooses different green development strategies to avoid blindly promoting industrial agglomeration, and prevent further deterioration of regional economic development. Besides, we need to keep a watchful eye on the adverse effects of resource misallocation on low-carbon development.

Finally, for the sake of promoting China's overall goal of achieving carbon reduction, other drivers need to be considered from a regional perspective. For the regions with high energy dependence, it is

necessary to improve energy efficiency through technological innovation, gradually change the energy consumption structure based on coal-based energy, and reduce the intensity of carbon emissions while also promoting the transformation of economic growth mode. For the eastern coastal areas with good economic conditions, it is necessary to upgrade the international competitiveness of new energy, develop a circular economy growth mode, and take a low-carbon, economical economic development route. In addition, it is worth noting that since China has not yet formed a unified carbon tax system, the state should establish a sound carbon tax mechanism, as soon as possible, and promote the construction of a carbon emissions trading mechanism to improve carbon trading sites and carbon trading platforms. In this process, the differential tax rate should be adopted in different regions and different industries.

Although this research has achieved some valuable results, it also has certain limitations. Other factors beyond the misallocation degree of resources may have threshold effects, and future research can continue to study by introducing other threshold factors, such as environmental regulation, industrial structure or regional development level. Furthermore, we also intrigued by the dynamic effects of misallocation, and there is resource misallocation across firms or industries, all of which require us to pay attention and will help us obtain more meaningful conclusions.

**Author Contributions:** Conceptualization, T.L. and D.H.; Methodology, D.H.; Software, D.H.; Validation, D.H. and S.F.; Formal Analysis, D.H.; Investigation, T.L. and D.H.; Resources, T.L. and D.H.; Data Curation, S.F. and L.L.; Writing—Original Draft Preparation, D.H.; Writing—Review and Editing, T.L., S.F. and L.L.; Visualization, D.H.; Supervision, T.L.; Project Administration, T.L.; Funding Acquisition, T.L., D.H., S.F. and L.L.

**Funding:** This research was funded by the Fundamental Scientific Research Funds for the Central Universities (3072019CFW0904).

**Acknowledgments:** We are indebted to the anonymous reviewers and editor.

**Conflicts of Interest:** The authors declare no conflicts of interest.

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
