# Peer review of "Can Industrial Co-Agglomeration between Producer Services and Manufacturing Reduce Carbon Intensity in China?"

_sustainability, doi:10.3390/su11154024_

Reviewer 1 Report

Dear Authors, I find the paper interesting for the academic debates, but I have several comments:
1. Section 3: Methodology and Data. Please, explain, why do you use this method?
2. You pay a lot of attention to «threshold effect». Please, explain in the paper, what is that
3. After literature review you jump in a lot of calculations. For the readers it might be not clear why you calculate this and that. Please, explain all your calculations clearer and their role in getting your research results
4. I would suggest to shorten the title of the paper

Author Response

Dear reviewer,

Thank you very much for your positive and constructive comments and suggestions on our manuscript (sustainability-550698). Those comments are valuable and very helpful for revising and improving our paper, as well as the important guiding significance to our research. We have revised the manuscript in detail according to your suggestions.In the attachment, we submitted a word file, in which the first half of the file is the modification description, and the second half is the revised version of this article.The responses to your comments are as follows:

Point 1: Methodology and Data. Please, explain, why do you use this method?  

Response 1: Thank you very much for your suggestion. Based on your suggestion, we reinterpreted why the threshold model was chosen in this paper. As stated in Section 5.1 of the paper, the reasons are as follows:

        From the Section 3 of the measurement of resource misallocation in various regions of China, it can be seen that the resource allocation situation in China is significantly different. Combined with the theoretical framework of the industrial synergy mechanism, we believe that when studying its carbon emission reduction effect, regional resource misallocation should be introduced into empirical analysis. we further speculate that the impact of industrial co-agglomeration on carbon intensity may have a typical “threshold effect”, that is, in the case of different economic parameters, the influence of independent variable on dependent variable may change. In this paper, when resource misallocation break through a certain threshold, the marginal coefficient of influence of industrial co-agglomeration on carbon intensity will become bigger or smaller, even the direction of action will be reversed.

   Hansen's nonlinear panel threshold regression model is a statistical analysis method that deals with nonlinear structural mutations. It can identify the data characteristics of unknown variables from the perspective of mathematical statistics, avoiding the bias caused by artificially dividing the threshold variable interval. Scientifically, a significant test was performed on the endogenous threshold effect. The empirical analysis of this method can not only clarify whether industrial co-agglomeration can help achieve carbon emission reduction targets, but also analysis the green energy conservation relationship dominated by industrial co-agglomeration based on the heterogeneity characteristics of different regions in China, namely the resource allocation status.

Point 2: You pay a lot of attention to «threshold effect». Please, explain in the paper, what is that?

Response 2: Thank you very much for your suggestion.

When an economic parameter reaches a certain value, it causes another economic parameter to suddenly affect other forms of development. The critical value at the root of this phenomenon is termed the threshold value, which is also known as the problem of nonlinear structural change, that is, threshold effect [1,2].

In this paper, we speculate that the effect of industrial co-agglomeration on carbon intensity may have a typical "threshold effect". That is to say, when the resource misallocation breaks through a certain threshold, the coefficient of influence of industrial co-agglomeration on carbon intensity will become larger or smaller, and even the direction of action will be reversed

Point 3: After literature review you jump in a lot of calculations. For the readers it might be not clear why you calculate this and that. Please, explain all your calculations clearer and their role in getting your research results. 

Response 3: Thank you very much for your suggestion.

Based on your suggestion, in order to enhance the clarity of the calculation of the article, and to facilitate readers to read and understand, we made the following changes.

On the one hand, we have described the calculation steps in more detail, and the results of the calculations are elaborated as much as possible.

For resource misallocation, the calculation steps are as following. First, we calculated the output elasticity of capital and labor (Section 3.1), then we calculated the misallocation index of capital and labor (Section 3.2), on the basis, we can measure the misallocation degree of resource (Section 3.3). Second, in Section 3.4, We analyze the results from the following aspects: First, there are resource misallocation in various regions of China, and there are large differences between regions. Secondly, in 2009, the misallocation of resource of China was more serious, and the central and western regions were more serious than the eastern regions, and we explained the reasons for the difference between regions. Third, in 2016, with the support of national policies, China’s resource misallocation has improved overall. However, the situation in individual provinces is still serious.

For carbon intensity, based on energy consumption, we measure the total amount of CO2 emissions according to the method provided by IPCC, and obtained the carbon intensity of each region in each year. According to the data of various regions in 2009, 2012 and 2016, the carbon intensity changes in China in recent years are analyzed. As a whole, China's carbon intensity is declining. From the perspective of regional distribution, the eastern region has a higher level of intensive economic development than the central and western regions. Combined with the reality of China's economic development, this paper has elaborated on this phenomenon. Finally, it explains in particular why Shanxi's carbon intensity has remained high.

On the other hand, at the end of the Section 1 and the beginning of Section 5.2, we introduced the reasons and the role of calculates: In section 3, first, we calculated the output elasticity of capital and labor (Section 3.1), then we calculated the misallocation index of capital and labor (Section 3.2), on this basis, we measured the misallocation degree of resource (Section 3.3), and has analyzed it comprehensively (Section 3.4). In section 4, the dependent variable, that is, the carbon intensity, is measured and analyzed. Section 5 presents threshold model and independent variable (industrial co-agglomeration) and control variables, besides, the threshold variable (resource misallocation) and the dependent variable (carbon intensity) are the results of the section 3 and section 4 above, respectively.

Point 4: I would suggest to shorten the title of the paper.

Response 4: Thank you very much for your suggestion. We changed the title to “Can industrial co-agglomeration between producer services and manufacturing reduce carbon intensity in China?”.

[1] Hansen, B.E. Threshold effects in non-dynamic panels: Estimation, testing, and inference[J]. Journal of Econometrics199993, 345-368.

[2] Hou, J.; Teo, T.S.H.; Zhou, F. Does industrial green transformation successfully facilitate a decrease in carbon intensity in China? An environmental regulation perspective. Journal of Cleaner Production 2018, 184, 1060-1071.

      Furthermore, we tried our best to improve the manuscript and made some changes in the manuscript. Nonetheless, these changes will not influence the content and framework of the paper.

        We appreciate for your warm work earnestly, and hope that the correction will meet with approval.

Thank you,

Best regards,

Dongri Han

Reviewer 2 Report

Dear Author,

The paper is well written and well structured. The subject analyzed is very interesting. Some tips to improve the paper:

- Introduction: it is necessary to make a more general introduction, considering the whole world (europe, united states) and not only china. Extending the problem increases the relevance of the paper. For example could be added this reference: 

A methodological approach to support the design of induction hobs, ASME International Mechanical Engineering Congress and Exposition, Proceedings (IMECE), 2016,

Investigating the feasibility of a reuse scenario for textile fibres recovered from end-of-life tyres, Waste Management 75, pp. 187-204

Comparative life cycle assessment of cooking appliances in Italian kitchens, Journal of Cleaner Production 186, pp. 430-449, 2018

Author Response

Dear reviewer,

Thank you very much for your positive and constructive comments and suggestions on our manuscript (sustainability-550698). Those comments are valuable and very helpful for revising and improving our paper, as well as the important guiding significance to our research. In the attachment, we submitted a word file, in which the first half of the file is the modification description, and the second half is the revised version of this article. The responses to your comments are as follows:

Point 1: The paper is well written and well structured. The subject analyzed is very interesting. Some tips to improve the paper:  Introduction: it is necessary to make a more general introduction, considering the whole world (europe, united states) and not only china. Extending the problem increases the relevance of the paper. For example could be added this reference:

1. A methodological approach to support the design of induction hobs, ASME International Mechanical Engineering Congress and Exposition, Proceedings (IMECE), 2016.

2. Investigating the feasibility of a reuse scenario for textile fibres recovered from end-of-life tyres, Waste Management 75, pp. 187-204.

3. Comparative life cycle assessment of cooking appliances in Italian kitchens, Journal of Cleaner Production 186, pp. 430-449, 2018.

Response 1: Thank you very much for your suggestion. On the basis of carefully reading the literatures you provided, we have rewritten the introduction of this paper, and extended the problem increases the relevance of the paper. Furthermore, we added references, that are, [1], [2] and [4], to this paper.

[1] Landi, D.; Cicconi, P.; Germani, M. et al. A Methodological Approach to Support the Design of Induction Hobs[C]// ASME International Mechanical Engineering Congress & Exposition, Proceedings (IMECE). 2016.

[2] Landi, D.; Gigli, S.; Germani, M. et al. Investigating the feasibility of a reuse scenario for textile fibres recovered from end-of-life tyres, Waste Management, 2018, 75, 187-204.

[4] Favi, C.; Germani, M.; Landi, D. et al. Comparative life cycle assessment of cooking appliances in Italian kitchens. Journal of Cleaner Production, 2018, 186, 430-449.

Furthermore, we tried our best to improve the manuscript and made some changes in the manuscript. Nonetheless, these changes will not influence the content and framework of the paper.

We appreciate for your warm work earnestly, and hope that the correction will meet with approval.